# The Intention to be Physically Active in Sedentary Obese Children: A Longitudinal Study

**DOI:** 10.3390/bs8010009

**Published:** 2018-01-11

**Authors:** Antonio García-Hermoso, Jose M. Saavedra, Yolanda Escalante, Ana M. Domínguez

**Affiliations:** 1Laboratorio de Ciencias de la Actividad Física, el Deporte y la Salud, Facultad de Ciencias Médicas, Universidad de Santiago de Chile (USACH), Santiago de Chile 81583799, Chile; antonio.garcia.h@usach.cl; 2Physical Activity, Physical Education, Health and Sport Research Centre (PAPESH), Sports Science Department, School of Science and Engineering, Reykjavik University, 101 Reykjavik, Iceland; 3Independent Researcher, 15179 Oleiros, Spain, yescgon@hotmail.com; 4Real Federación Española de Salvamento y Socorrismo, 28703 Madrid, Spain; adominguezpachon@hotmail.com

**Keywords:** adherence, obesity, children, physical activity

## Abstract

Obese children are usually less active than their normal-weight counterparts, although the reasons for this remain unclear. The objective of the present study was to determine how a long-term program (3 years of intervention and 6 months of follow-up detraining) of physical exercise with or without a low calorie diet influenced sedentary obese children’s intention to be physically active. The participants were 27 children, ages from 8 to 11 years, who formed two groups according to the program that they followed. One group followed an exercise program (three 90-min sessions per week), and the other this same exercise program together with a hypocaloric diet. The intention to be physically active was assessed via the Measurement of Intention to be Physically Active (MIFA) questionnaire. The subjects’ scores at different times of the program (baseline, Year 3, and detraining) were compared using a repeated-measures ANOVA, and a post-hoc Tukey’s test was applied to confirm the differences. After both the intervention and detraining, both groups showed greater intention to be physically active. This suggests the suitability of long-term physical exercise to generate greater intention to be physically active and thus establish healthy life habits including increased levels of physical activity.

## 1. Introduction

Obesity is a major public health problem and has become epidemic [1] as evidenced by the rapid increase in its prevalence worldwide, especially in children. In developed countries, the prevalence of overweight and obesity has increased to 23.8% in boys and 22.6% in girls [2]. Obesity in childhood has negative consequences for physical and psychological health [3]. Governments internationally are therefore working on strategies for its prevention [4]. For weight control in childhood, these strategies generally focus on increasing physical activity (PA) and/or decreasing sedentary behavior [5] in recognition of the important role that PA can play in preventing overweight and obesity [6].

The World Health Organization recommends that children and adolescents (5–17 years old) should accumulate a daily minimum of 60 min of mostly aerobic PA of moderate or vigorous intensity. A minimum of three times a week is also recommended for the practice of activities that strengthen the musculoskeletal system [1]. However, it is worrying that the international Health Behavior in School-Aged Children study reports that those recommendations are met by only 27% of girls and 40% of boys [1], and, in Spain in particular, by fewer than 50% of children and adolescents [7].

Obese children appear to be less physically active than their normal-weight counterparts, although the reasons remain unclear [8]. Studies on the subject do indicate, however, that the overweight and obese report a less positive attitude towards PA, and perceive more barriers and fewer benefits than their normal-weight counterparts [9,10]. A recent study with obese children reported that long-term physical exercise enhances their autonomous motivation, and hence favors changes in them towards healthy living habits that are stable over time [11]. According to the Theory of Reasoned Action, a person’s attitude towards PA will be an important predictor of its actual performance [12]. Obese children themselves generally point to their weight problem being the main barrier to their doing PA [10,11]. The authors of these last two studies suggest that interventions designed to increase activity levels in overweight and obese adolescents should focus on reducing barriers that are related to weight, with separate exercise sessions organized for young people with this problem [10]. The objective of the present study was therefore to determine how a long-term program of physical exercise (3 years of intervention and 6 months of follow-up detraining) associated or not with a low calorie diet influenced sedentary obese children’s intention to be physically active.

## 2. Materials and Methods

### 2.1. Participants

A total of 135 subjects were invited to participate through the collaboration of various schools in the town of Cáceres (Spain). The inclusion criteria were (i) a body mass index (BMI) at or above the 97th percentile for their age and sex according to the curves of the Spanish school population [13], (ii) an age from 8 to 11 years, (iii) being sedentary, and (iv) not undergoing any treatment or intervention for their condition. Subjects were excluded if they (i) were regularly practicing PA, or following an exercise program or some other therapy (*n* = 65), (ii) were involved in any weight control program (*n* = 18), (iii) were taking any medication (*n* = 8), (iv) had any type of dysfunction limiting their PA (*n* = 2), or (v) had any other reasons that would preclude them from involvement (*n* = 15). The resulting sample consisted of 27 sedentary Caucasian children (ages 10.4 ± 1.0 years). They formed two groups (according to whether or not they were in the school meal program): the E group who followed a multi-sports exercise program alone (*n* = 11, 8 boys and 3 girls), and the E + D group who followed a combination of this program plus a low calorie diet (*n* = 16, 10 boys and 6 girls) (Figure 1). All the children’s parents completed a prior informed consent form. The study was approved by the Bioethics and Biosafety Committee of the University of Extremadura, and respected the principles of the Declaration of Helsinki.

### 2.2. Physical Exercise Program

The exercise program consisted of three 90 min sessions per week for a period of three years. The total number of sessions was 230, and the total duration of the program 20,700 min. The sessions took place in a sports hall, under the supervision of two doctoral students in Physical Activity and Sports Sciences (A.G-H., A.M.D.) under the supervision of two PhD’s in Physical Activity and Sports Sciences (J.M.S., Y.E.). The sessions consisted of a warm-up (15–20 min), a main part of multi-sport games and activities with a predominantly aerobic component (moderate to vigorous intensity) (60–65 min) combined with strength work, and a cool-down (5–10 min). In as far as possible, the tastes and sporting interests of the participants were respected, giving them different activities to choose from in each session, encouraging cooperative activities and interpersonal relationships. The intensity of the session was monitored by accelerometry to ensure that all the subjects performed the activities with the same intensity. The accelerometer used was a Caltrac (Hemokinetics, Madison, WI, USA), programmed to function as a PA monitor. This uni-axial accelerometer contains a piezoelectric bender element, which assesses the intensity of movement in the vertical plane (“motion counts”). Its validity has been demonstrated as a method for estimating energy expenditure in children [14], and it has been used in other studies [11,15,16]. A Caltrac accelerometer does not record such activities as rowing or swimming. However, no activity of this type was used either in the exercise program or in the subjects’ recorded daily PA. This accelerometer enabled a rising progression of the intensity of exercise over the three years of intervention to be established.

### 2.3. Diet Program

The low-calorie diet consisted of five balanced meals spread throughout the day, with an energy intake of 1500 kcal/day. Some studies have recommended diets of between 1500 and 1800 kcal/day in obese children who are still growing, since in this way their growth and development are not compromised [17]. Thus, the diet prescribed was of 1500 kcal/day, similar to that of other studies [18]. The diet consisted of 57% carbohydrates, 17% proteins, and 26% fats. Foods were selected according to the subject’s dietary habits. The subjects were given a list of food groups (vegetables, fruit, meat, fish, eggs, cereals, legumes, and dairy) and their cooking methods, the weekly consumption frequencies, and maximum daily amounts. They chose the foods from each group according to the indications they were given and to their own particular tastes. General recommendations were established focused on basic healthy lifestyle eating: consume ≥5 servings of fruits and vegetables every day; minimize sugar sweetened beverages such as soft drinks, sports drinks, and sugar-added fruit juices; have more meals prepared at home rather than purchasing take-away restaurant food. Regular meetings were held with the children’s parents for the control and monitoring of the diet.

### 2.4. Measures

At baseline, Year 3, and six months after termination of the program (detraining), all the subjects underwent the following assessments: pubertal status, eating habits, daily PA, kinanthropometry, and intention to be physically active. The pubertal status of the children was assessed by the Head of the Pædiatric Hospital San Pedro de Alcántara from the Tanner stages of pubic hair development [19]. Eating habits were assessed by means of an ad hoc questionnaire covering three days (Thursday, Friday, and Saturday) completed by the parents. The weight of food was estimated from the register made by the parents. The average of the three days was taken (kcal/day) and recorded. The NutrIber computer program was used to calculate the daily food intake.

Daily PA was assessed each year using a validated uniaxial accelerometer (Caltrac) over three consecutive days (Thursday, Friday, Saturday), except for bath or shower time. All the participants were instructed on how to record the amount of time spent cycling or swimming during the evaluation period. The subjects recorded the number of accelerometer “motion counts” at the beginning and the end of the day [16]. The final Caltrac score was taken as the average of the three days (motion counts/day).

The kinanthropometric measurements were carried out following the ISAK protocol for height, weight, fat mass, and lean mass (bioimpedance) using a standard scale and stadiometer (Seca, Berlin, Germany) and bioimpedance meter (Bodystat^®^ 1500, Bodystat-USA Inc., Tampa, FL, USA). The body mass index (BMI) was calculated as weight divided by height squared (kg/m^2^), and the standardized deviations, zBMI, were then computed based on the Spanish population [13].

### 2.5. Intention to be Physically Active

The MIFA (Measurement of Intention to be Physically Active) questionnaire self-administered questionnaire consists of five items designed to measure the subject’s intention to be physically active after passing through the various levels of education (primary school, secondary school). The original scale was published by Hein et al. [20]. Its adaptation and validation is due to Moreno et al. [21]. The items are preceded by the phrase “Regarding your intention to practice some physical/sporting activity: …”. The five items are as follows: (a) I am interested in developing my fitness; (b) Apart from the physical education classes, I like doing sports; (c) After finishing secondary school, I would like to be part of a sports training club; (d) After finishing secondary school, I would like to keep physically active, and (e) In my spare time, I usually practice some sport. The responses were on a 5-point Likert scale, where 1 corresponded to “strongly disagree” and 5 to “strongly agree.”

### 2.6. Data Analysis

All variables satisfied the Kolmogorov–Smirnov normality and Levene homoskedasticity conditions. The basic descriptive statistics (mean and standard deviation) were calculated. A one-way ANOVA was performed to ensure that there were no differences between the groups before intervention in age, maturational development, energy intake, and anthropometric and body-composition parameters. A repeated-measure ANOVA and a post-hoc Tukey test were applied to compare the interactions between the different assessment times (baseline, Year 3, and detraining) and the two groups (exercise only, E group; exercise plus hypocaloric diet, E + D group). The significance level for all statistical tests was set at *p* ≤ 0.05. All calculations were performed using the SPSS version 16.0 software package.

## 3. Results

Table 1 lists the characteristics of the two groups. Before intervention, there were no significant differences in any of the variables studied. Adherence to the exercise program was evaluated as the percentage of attendance at the sessions. For the three years of the program, this adhesion was greater than 78%. The intensity of 39 randomly selected sessions (13 sessions per year) was quantified, allowing an objective determination that all the subjects carried out the same exercise program and that there was a steady progression over the three years of the intervention. Not all sessions were quantified, since the time required to program and put in place each accelerometer necessarily had to be subtracted from the core exercise time.

Table 2 lists the changes in the intention to be physically active (MIFA) and the daily physical activity at different assessment times (baseline [a], Year 3 [b], and detraining [c]) in the two groups, E group and E + D group. The intervention led to longitudinal changes in the intention to be physically active (a < b): E group (*p* < 0.001) and E + D group (*p* = 0.001). Similarly, after the detraining period, there was still an increase in this parameter (a < c) in both groups: E group (*p* = 0.001) and E + D group (*p* < 0.001). Finally, the daily PA of both groups had increased significantly after the detraining period relative to the baseline, reflecting a possible establishment of healthy habits after both kinds of treatment. There were no differences between the two groups either in the intention to be physically active or in daily PA (*p* > 0.05) at any of the three times of the measurements (Table 2).

## 4. Discussion

This longitudinal intervention (three years program and six months follow-up/detraining) was based on a program of physical exercise with or without diet, analyzing the influence on obese children’s intention to be physically active. The results indicated that both of these long-term interventions favored this intention, and that this effect was maintained over time after the intervention. No differences were observed between the two interventions, so that the exercise itself appears to be beneficial in this regard.

With respect to the intragroup results, the analyses showed an increase in the subjects’ intention to be physically active after 3 years of intervention in the E group (2.33 ± 1.01 vs. 4.34 ± 0.40, *p* < 0.001) and the E + D group (2.82 ± 0.81 vs. 4.32 ± 0.71, *p* < 0.001), and after detraining in the E group (2.33 ± 1.01 vs. 4.66 ± 0.47, *p* < 0.001) and the E + D group (2.82 ± 0.81 vs. 4.51 ± 0.62, *p* = 0.001). The study has shown that physical exercise with and without dieting can generate healthy habits even after the intervention has been discontinued. Initially the subjects were sedentary and hence showed little intentionality to be physically active. Since obese children perceive greater barriers to physical activity (related to their weight) and fewer benefits [10], the exercise program focused on providing such children with enjoyable activities with which they could experience success. Likewise, they were encouraged verbally to participate in different exercises (“You can do it”), and their potential anxiety associated with participating in PA was reduced by proposing activities that were primarily cooperative [22]. Thus, it seems appropriate to focus on reducing the barriers related to the subjects’ weight in order to foster their enjoyment of exercise [9,10]. In this sense, it is of paramount importance that each subject find the physical exercise that is most pleasing to them, since in this way one could be sure of stricter adherence to the exercise program [23]. In addition to finding the preferred physical exercise, the intention to be physically active could be enhanced with such actions as walking or cycling to school. These strategies not only have an impact on health, but also on academic performance [24].

These longitudinal programs have the capacity of offering a variety of activities, exercises, and sports. The obese subjects would thus have a wider choice of activities to perform, and hence perceive fewer barriers to performing PA—they usually choose the activity in which they perceive themselves to be competent [9,11]. In fact, perceived competence seems to be one of the strongest predictors of intention to be physically active [21]. In analyzing the daily PA results, we found these obese subjects to have increased their PA relative to level at the start of the program (this was so immediately after the intervention and after detraining). This could be a reflection of the generation of healthy lifestyles, confirming the subjects’ greater intention to be physically active. Longitudinal physical-exercise-based interventions independently encourage subsequent adherence to exercise after the program has finished. In general, obese children are commonly dissatisfied with their bodies [25] and that this is particularly salient in overweight/obese girls [26]. As the program progresses and they lose weight, this perception might change, which would facilitate their adherence to exercise since there is a direct inverse relationship between satisfaction with one’s body and adherence to PA [27].

The intergroup analysis showed no differences between the two groups in the benefits with respect to the intention to be physically active. Studies in this regard in adults indicate that an initial focus on diet is associated with short-term weight loss, while a focus on factors related to PA, with particular emphasis on sources of intrinsic motivation, play a more important role in long-term weight control [28]. The present results are this further support of the suitability of exercise-based programs to favor weight control and a higher level of PA. This could be because following a hypocaloric diet requires a great deal of effort (every day of the week, five times a day), especially if the subject is eating away from home.

A number of limitations of this study should be taken into consideration. Firstly, there was no control group with which to compare the results and determine the effectiveness of the exercise program. Clearly, however, since obesity is a disease, it is difficult to obtain a sample of children who do not undergo treatment for their condition. Secondly, the number of participants was small (*n* = 27), although the longitudinal nature of the study could lessen the importance of this limitation.

In summary, the present study has shown how physical exercise and diet influence the creation of healthy habits in obese children. The findings suggest that, after longitudinal exercise programs with and without diet, obese children increase their intention to be physically active, with increased levels of daily PA not just immediately following the program but also after a period of detraining. This seems to be indicative of the suitability of exercise adapted to the obese population in generating and maintaining a greater intention to be physically active.

## Figures and Tables

**Figure 1 behavsci-08-00009-f001:**
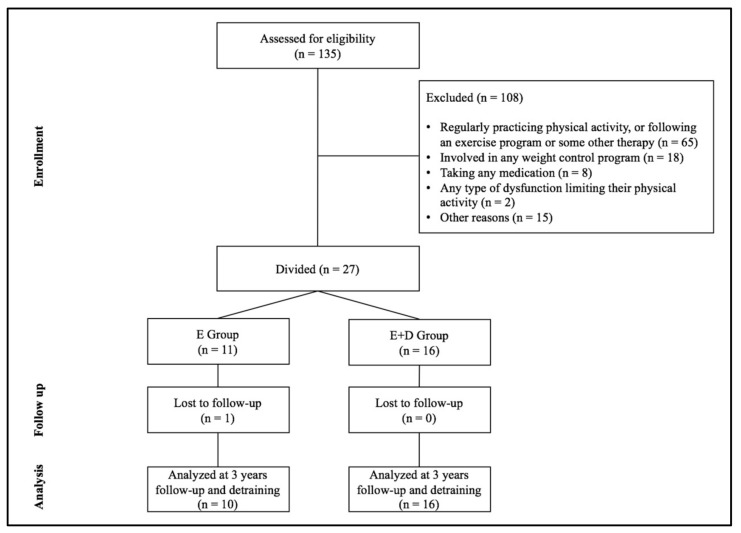
Participant selection of study.

**Table 1 behavsci-08-00009-t001:** Characteristics of the sample before the intervention, including the means and standard deviations, and the values of the one-way ANOVA *F* and *p* statistics.

	E Group (M ± SD)	E + D Group (M ± SD)	*F*	*p*
*n* (boys/girls)	11 (8/3)	16 (10/6)	-	-
Age (years)	10.73 ± 0.90	10.19 ± 1.02	2.630	0.085
Tanner stage (pubic hair)	2.01 ± 0.63	1.81 ± 0.54	0.068	0.796
Eating habits (kcal/day)	1912.96 ± 204.10	1906.26 ± 210.92	0.017	0.897
Height (m)	1.49 ± 0.10	1.46 ± 0.10	0.530	0.473
Weight (kg)	62.15 ± 9.86	57.68 ± 11.20	0.257	0.617
BMI (kg/m^2^)	27.85 ± 3.42	27.32 ± 3.88	0.507	0.483
Fat mass (%)	25.47 ± 6.90	25.89 ± 5.71	0.376	0.545

M: mean; SD: standard deviation; E: exercise group; E + D: exercise plus diet group.

**Table 2 behavsci-08-00009-t002:** Changes in intention to be physically active (MIFA) and daily physical activity (PA) at different times of assessment, including the means and standard deviations, and the values of the repeated measures ANOVA *F* and *p* statistics.

			Intervention		Detraining	ANOVA		
			Baseline	3rd Year	6 Months			
			M ± SD	M ± SD	M ± SD			
		*n*	a	b	c	*F*	*p*	Diff.
MIFA	E	10	2.33 ± 1.01	4.34 ± 0.40	4.66 ± 0.47	26.154	<0.001	a < b, c
	E + D	16	2.82 ± 0.81	4.32 ± 0.71	4.51 ± 0.62	11.553	<0.001	a < b, c
	*p*		0.232	0.954	0.636			
Daily PA	E	10	443.13 ± 106.45	611.81 ± 159.29	724.30 ± 124.33	10.185	0.001	a < b, c
	E + D	6	442.65 ± 99.56	526.18 ± 120.97	771.49 ± 91.84	23.966	<0.001	a < b, c
	*p*		0.981	0.259	0.439			

Measurement of Intention to be Physically Active (MIFA) questionnaire; PA: Physical activity; M: mean; SD: standard deviation; E: exercise group; E + D: exercise plus diet group; a: baseline; b: 3rd year; c: after 6 month interventions.

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
