# Peer review of "The Intention to be Physically Active in Sedentary Obese Children: A Longitudinal Study"

_behavsci, 2018, doi:10.3390/bs8010009_

Round 1

Reviewer 1 Report

A valuable paper which would be of interest to readers. However, the study is limited by the very small sample size. The paper is poorly structured at present with results in the methods section – please amend so that results are contained in the results section only. A participant flow chart would be of value in the paper as it is difficult for a reader to assess study quality on the basis of this paper; eg dropout/loss to follow-up. Abstract Line 17 – associated or nut – suggest to change to with and without Introduction Line 31 – suggest to include prevalence of child obesity rather than obesity. Reference for the opening sentence should be a primary reference rather than a report from WHO (ref 1) Line 38 – suggest to change youngsters to adolescents or youth rather than youngsters Materials and methods Results are currently present in this section and should instead be in the results section – eg age of participants, table 1 Formatting in table 1- number of decimal places to which data are reported varies, please change to be consistent Table 1 could also contain more data which are presented in text, eg n for each group, gender split. There is confusion around the use of G1 G2 to describe group as well as E+D and E groups – please make this much clearer. The paper describes two groups but I can’t work out where they are from; E n=11; E+D n=16; G1 n=8; G2 n=10 ? are G1 and G2 the completers? This needs to be clearer and I suggest a flow chart. Table 1 is without footnotes which would allow it to stand alone, eg what is F and what test has been performed, if eating habits is total energy intake, then please state this rather than total energy. Why is fat mass expressed as % and lean mass as kg? should these also be consistent for comparison? I also question the SD for BMIz in both groups considering inclusion criteria were for children above the 97th percentile. SD seems very very large in both groups and mean BMIz does appear to differ by 0.8 BMIz units Line 111-112 needs elaboration and further explanation Foods were selected according to the subject's dietary habits.  Line 116 – avoid the use of etc, instead say including x,y,z. Line 137 – suggest to include the name of the reference population used for BMIz Line 152 – delete it The present intervention it is a quasi-experimental study Results Start the results section with demographics of the population and whether there were any differences between groups Also include data on retention and attendance here rather than in the methods section Please include footnotes for table 2 to explain abbreviations so that table stands alone. Lines 168-170 Delete the text This 168 section may be divided by subheadings. It should provide a concise and precise description of the 169 experimental results, their interpretation as well as the experimental conclusions that can be drawn.  please include n values in all tables - are the data complete? Discussion includes too much data which should be in the results. Please change discussion to summarise key findings generally and not to the level of providing data.

Author Response

Reviewer #1

Dear Reviewer #1

Thank you very much for your comments. We respond point by point below.

Reviewer #1

A valuable paper which would be of interest to readers. However, the study is limited by the very small sample size.

Authors: You are right. However, the amount of intervention in the sample is relevant: 3 years, 270 sessions of 90 minutes.

Reviewer #1

The paper is poorly structured at present with results in the methods section – please amend so that results are contained in the results section only.

Authors: Done.

Reviewer #1

A participant flow chart would be of value in the paper as it is difficult for a reader to assess study quality on the basis of this paper; eg dropout/loss to follow-up.

Authors: Done.

Reviewer #1

Abstract Line 17 – associated or nut – suggest to change to with and without

Authors: Done.

Reviewer #1

Introduction

Line 31 – suggest to include prevalence of child obesity rather than obesity.

Authors: Done.

Reviewer #1

Reference for the opening sentence should be a primary reference rather than a report from WHO (ref 1)

Authors: Done.

Reviewer #1

Line 38 – suggest to change youngsters to adolescents or youth rather than youngsters

Authors: Done.

Reviewer #1

Materials and methods Results are currently present in this section and should instead be in the results section – eg age of participants, table 1 Formatting in table 1- number of decimal places to which data are reported varies, please change to be consistent Table 1 could also contain more data which are presented in text, eg n for each group, gender split.

Authors: Table 1 has been passed to the Results section and modified following all your indications. A sentence has also been added to the Data Analysis section on the analysis made in this table.

Reviewer #1

There is confusion around the use of G1 G2 to describe group as well as E+D and E groups – please make this much clearer.

Authors: The terminology has been homogenized, always using E group and E+D group.

Reviewer #1

The paper describes two groups but I can’t work out where they are from; E n=11; E+D n=16; G1 n=8; G2 n=10 ? are G1 and G2 the completers? This needs to be clearer and I suggest a flow chart.

Authors: A flow chart has been added. The number of subjects invited to participate in the study and the reasons for exclusion have also been added to the Participants section.

Reviewer #1

Table 1 is without footnotes which would allow it to stand alone, eg what is F and what test has been performed, if eating habits is total energy intake, then please state this rather than total energy.

Authors: Table 1 has been moved to the Results section and modified in accordance with your indications.

Reviewer #1

Why is fat mass expressed as % and lean mass as kg? should these also be consistent for comparison?

Authors: As for the previous comment, Table 1 has been moved to the Results section and modified in accordance with your indications.

Reviewer #1

I also question the SD for BMIz in both groups considering inclusion criteria were for children above the 97th percentile. SD seems very very large in both groups and mean BMIz does appear to differ by 0.8 BMIz units.

Authors: As for the previous comments, Table 1 has been moved to the Results section and modified in accordance with your indications.

Reviewer #1

Line 111-112 needs elaboration and further explanation Foods were selected according to the subject's dietary habits. 

Authors: Clarified.

Reviewer #1

Line 116 – avoid the use of etc, instead say including x,y,z.

Authors: Done.

Reviewer #1

Line 137 – suggest to include the name of the reference population used for BMIz

Authors: Done.

Reviewer #1

Line 152 – delete it The present intervention it is a quasi-experimental study

Authors: Done.

Reviewer #1

Results

Start the results section with demographics of the population and whether there were any differences between groups Also include data on retention and attendance here rather than in the methods section

Authors: Done.

Reviewer #1

Please include footnotes for table 2 to explain abbreviations so that table stands alone.

Authors: Done.

Reviewer #1

Lines 168-170 Delete the text This 168 section may be divided by subheadings. It should provide a concise and precise description of the 169 experimental results, their interpretation as well as the experimental conclusions that can be drawn.  please include n values in all tables - are the data complete?

Authors: Deleted.  Values of n added to the table.

Reviewer #1

Discussion includes too much data which should be in the results. Please change discussion to summarise key findings generally and not to the level of providing data.

Authors: We have revised these two sections carefully, and added sentences in accordance with your indications.

Finally, we sincerely appreciate the time you spent in reviewing this work. Likewise, we appreciate your comments and suggestions, which we believe have improved the quality of the paper.

We trust that the changes made will be sufficient to recommend acceptance of the work.

Thank you again.

The Authors

Reviewer 2 Report

Needs proof reading to correct grammar and spelling (examples: lines, 51, 52 and 53)

In general the purpose of the study is of scientific relevance but the choice of grammar used makes it hard to follow.

Author Response

Reviewer #2

Needs proof reading to correct grammar and spelling (examples: lines, 51, 52 and 53)

In general the purpose of the study is of scientific relevance but the choice of grammar used makes it hard to follow.

Dear Reviewer #2

Thank you very much for your remarks on our paper.  The grammar and syntax of the entire manuscript has been checked carefully and corrected by an English Ph.D. (with an M.A. from Cambridge University, and a Ph.D. from Florida State University), and with whom we have often worked.

We trust that the changes made will be sufficient to recommend acceptance of the work.

Thank you again.

The Authors

Round 2

Reviewer 1 Report

changes have improved the manuscript quality but there is still a demonstrated lack of attention to detail.

specific comments:

Figure 1 - this is a good figure to have in the manuscript. suggest to change figure caption to Flow chart of study (participants or participant selection) as it is not a flow chart of the study.

page 3, line 110 - "also, a of general" - please amend

line 114 and line 141 - use of etc. please remove as per comment in first review

line 161 - 162 - please change no differences to no significant differences

Table 1 - inconsistency in reporting of data to decimal places - must amend to be standardised. Table 1 also includes footnote for abbreviation for mean as M, but this is not used - remove

Table 2 does not include key abbreviations as footnotes - MIFA, a, b, c - it is difficult to follow without substantial reference to the text. Tables should stand alone.

line 211 - "to all this there has to be added that" - not scientific, suggest delete or re-word

line 213-14 "obese children are commonly dissatisfied with their bodies" - this absolutely needs a reference and needs to be refined - are all obese children dissatisfied with their bodies or are there perhaps differences between age groups and gender?

Author Response

Reviewer #1

Comments and Suggestions for Authors

changes have improved the manuscript quality but there is still a demonstrated lack of attention to detail.

Authors:

Dear Reviewer #1

Thank you very much for your positive comments. We have put more effort on the details. We respond point by point below.

Reviewer #1

specific comments:

Reviewer #1

Figure 1 - this is a good figure to have in the manuscript. suggest to change figure caption to Flow chart of study (participants or participant selection) as it is not a flow chart of the study.

Authors: Done

Reviewer #1

page 3, line 110 - "also, a of general" - please amend

Authors: Done

Reviewer #1

line 114 and line 141 - use of etc. please remove as per comment in first review

Authors: Done

Reviewer #1

line 161 - 162 - please change no differences to no significant differences

Authors: Done

Reviewer #1

Table 1 - inconsistency in reporting of data to decimal places - must amend to be standardised. Table 1 also includes footnote for abbreviation for mean as M, but this is not used – remove

Authors: Done. We have just used two decimals for mean and standard deviation and three decimals for F and p-value.

Reviewer #1

Table 2 does not include key abbreviations as footnotes - MIFA, a, b, c - it is difficult to follow without substantial reference to the text. Tables should stand alone.

Authors: We have just used two decimals for mean and standard deviation and three decimals for F and p-value. Abbreviations were included as footnotes

Reviewer #1

line 211 - "to all this there has to be added that" - not scientific, suggest delete or re-word

Authors: Done

Reviewer #1

line 213-14 "obese children are commonly dissatisfied with their bodies" - this absolutely needs a reference and needs to be refined - are all obese children dissatisfied with their bodies or are there perhaps differences between age groups and gender?

Authors: References have been added.

Finally, we sincerely appreciate the time you spent in reviewing this work. Likewise, we appreciate your comments and suggestions, which we believe have improved the quality of the paper.

We trust that the changes made will be sufficient to recommend acceptance of the work.

Thank you again.

The Authors